# Monocyte clusters suggestive of a chronic inflammatory phenotype are associated with reduced endothelial function in Veterans with respiratory symptoms

Gregory Pappas[1], Carol Gardner[2], Changjiang Guo[2], Kathleen Black[3], Raymond Rancourt[3], Debra Laskin[2], Howard Kipen[3], Anays Sotolongo[1,4], Michael Falvo[1,4], Andrew Gow[2]*

1 Airborne Hazards and Burn Pits Center of Excellence, Veterans Affairs New Jersey Health Care System, East Orange, New Jersey, United States of America, 2 Ernest Mario School of Pharmacy, Rutgers University, Piscataway, New Jersey, United States of America, 3 Environmental and Occupational Health Sciences Institute, Rutgers University, Piscataway, New Jersey, United States of America, 4 New Jersey Medical School, Rutgers University, Newark, New Jersey, United States of America

* gow@research.rutgers.edu

## Abstract

Exposure to airborne hazards during deployment is associated with persistent respiratory symptoms among military veterans even years after deployment. Circulating monocytes, key components of the innate immune response, are implicated in inflammatory processes that may be sustained long after such exposures and contribute to related health issues. This cross-sectional study, conducted years after deployment, aimed to characterize monocyte activation profiles in veterans with deployment-related respiratory symptoms and investigate associations with physiological markers of pulmonary and vascular function. Circulating monocyte immunophenotype, pulmonary function, and brachial artery flow-mediated dilation (FMD) were assessed in 82 previously deployed veterans. Using principal component and hierarchical clustering analyses, we identified two distinct monocyte activation phenotypes: Cluster 1, characterized by elevated CD87, CD11b, and CD163, and cluster 2 which expressed markers of non-classical monocytes and CD195, indicative of a chronic inflammatory phenotype. Veterans in cluster 2 exhibited impaired endothelial-dependent vasodilation (FMD/NMD ratio; p = 0.02) and elevated airway resistance (R5; p = 0.01), despite normal pulmonary function. These findings suggest an association between distinct monocyte activation profiles and measures of microvascular and airway dysfunction in this cohort, potentially reflecting sustained inflammation secondary to environmental exposure. These observed associations underscore the need for further research into the role of monocytes in these long-term physiological changes. Elucidating the mechanistic pathways by which these monocyte phenotypes may contribute to persistent physiological alterations is critical and could inform future strategies for

**Data availability statement:** Data were collected under a HIPPA authorization that did not include a provision to share data publicly. However, these data will be placed into a data repository whereby investigators with research protocols approved by their institution may apply to receive data pursuant to data use and transfer agreements. Inquires into this process may be sent to the VA's Airborne Hazards and Burn Pits Center of Excellence (vhaeasairhazardscoe@va.gov).

**Funding:** This work was supported by Merit Review Award # I01 CX001515 from the United States (U.S.) Department of Veterans Affairs Clinical Sciences Research and Development Service and supported in part by the Airborne Hazards and Burn Pits Center of Excellence (Contract No. 36C24519C0225). The contents do not represent the views of the U.S. Department of Veterans Affairs or the United States Government.

**Competing interests:** The authors have declared that no competing interests exist.

identifying at-risk veterans or exploring novel immunomodulatory approaches if such links are further substantiated.

## Introduction

Inhalable levels of fine particulate matter (PM) air pollution are ubiquitous in the Southwest Asia Theater of Military Operations and at levels approximately 10-fold greater than those observed in urban U.S. cities [1,2]. Anthropogenic (e.g., smoke from open burn pits) and regional sources (e.g., dust and sand) of airborne hazards contribute to high PM levels and the complex and variable composition of these airborne particulates.. Exposure to these airborne hazards is associated with post-deployment respiratory symptoms of dyspnea, cough and wheeze [3], and may contribute to deployment-related respiratory disease [4]. Inhalation of particulate matter (PM) is known to activate cell-signaling pathways leading to pro-inflammatory responses [5,6]. Such inflammatory mechanisms are hypothesized to contribute to the development or persistence of post-deployment respiratory symptoms and conditions. This type of response is consistent with the well-established civilian air pollution literature [7–10] and provides a target for investigation in deployed military personnel.

Circulating monocytes serve as a likely target as they are key components of innate immune responses; moreover, subsets (i.e., classical, non-classical, and intermediate) of these cells exhibit functions, relevant to disease pathogenesis [11,12]. An understanding of circulating monocyte subsets and their activation status may also be of relevance in deployed military personnel with airborne hazards exposure due to their role in both initiating and *sustaining* inflammatory processes. To our knowledge, monocyte behavior has not been investigated in these individuals and this represents the focus of our studies. Our primary objectives were to: 1) characterize the monocytic profile of deployed military veterans, and 2) examine the associations between monocyte surface marker expression and reported symptoms, as well as biochemical and physiological markers of endothelial and pulmonary function. We speculate that monocytes displaying higher levels of known activation markers will be correlated with indicators of compromised vascular health, including impaired endothelial-dependent vasodilation, reduced pulmonary diffusing capacity, and increased self-reported dyspnea. Understanding the inflammatory profiles of monocytes in this population could provide insights into both the pathogenesis of deployment-related respiratory disease and associated health outcomes Such insights may help identify novel biological markers or illuminate pathways that could, if further research establishes clearer mechanistic roles, inform future therapeutic considerations.

## Methods

### Study participants and procedures

Military veterans (n = 84) deployed to the Southwest Asia Theater of Military Operations after 2001 were enrolled in a parent study examining deployment-related airborne hazards and respiratory symptoms. The enrollment period was from January

8th 2019 until April 15th 2024. To be eligible for the study, participants were required to have no history of pulmonary disease, cardiovascular disease, diabetes, cancer (excluding non-melanoma skin cancer), uncontrolled hypertension, severe psychiatric illness, or a current smoker or pregnancy. Individuals with conditions necessitating systemic steroids or other immunomodulators that could significantly affect our outcomes were typically excluded based on their underlying diagnoses. Testing was conducted over multiple laboratory visits and included: i) medical and military history review, ii) pulmonary function testing (PFT), iii) flow- and nitroglycerine-mediated dilation (FMD, NMD) of the brachial artery, and iv) venous blood for cell isolation and immunophenotyping. All procedures were reviewed and approved by the VA New Jersey Health Care System Institutional Review Board (IRB) and Rutgers University IRB, and all participants provided written informed consent.

## Self-reported symptoms and history

All participants completed a series of electronic surveys using a tablet or computer, and data were managed via the secure, web-based software application REDCap (Research Electronic Data Capture) hosted at the East Orange VA Medical Center [13,14]. Surveys collected information from participants' medical, health, and military history. Additionally, participants were asked to characterize their deployment-related exposures. Included questions mirrored those reported in the VA's Airborne Hazards and Open Burn Pit Registry self-assessment questionnaire [15]. Topics assessed ranged from deployment history (i.e., deployment or region related exposures), medical history or symptoms (i.e., NHIS Functional Limitation Scale, the Modified Medical Research Council (mMRC) Dyspnea Scale, general health concerns, alcohol and tobacco use, and other occupational exposures. Additional questions including the St. George Respiratory Questionnaire (SGRQ) focused on the impact of respiratory symptoms on overall health, daily life, and perceived well-being.

For analysis, total length of deployment was calculated as total time deployed across one or multiple deployments, and time since first deployment was calculated as the difference between the date of the first study visit and end of the first deployment. Responses to the NHIS Functional Limitation Scale were grouped into four categories: 1) No difficulty, 2) Some difficulty (collapsed from 'a little' and 'somewhat' difficult), 3) A lot of difficulty (collapsed from 'very difficult' and 'cannot perform'), or 4) Unknown/Do not perform. A total score of zero or one on the mMRC was grouped as 'no dyspnea in daily living', while a score of two or higher was defined as 'dyspnea with daily living'.

## Pulmonary function

Testing included spirometry and the diffusing capacity of carbon monoxide ($DL_{CO}$). Spirometry was performed both pre-and post-bronchodilator administration (4 puffs of albuterol via metered dose inhaler). $DL_{CO}$ was assessed via the single-breath technique, with appropriate corrections for hemoglobin concentration. All testing was performed by trained technicians following previously published guidelines with commercially available equipment (Cosmed, Quark PFT/Q-Box/i2M; Rome, Italy) [16]. Predicted values for spirometry and $DL_{CO}$ based on age, race, and gender were calculated from the Global Lung Function Initiative calculator from the European Respiratory Society [17]. Spirometry variables of interest included forced vital capacity (FVC), forced expiratory volume in one second ($FEV_1$), and the $FEV_1/FVC$ ratio.

Subjects also completed respiratory oscillometry (FOT), a non-invasive measure of respiratory mechanics assessed by the impedance of the respiratory system during tidal breathing. FOT was performed using optimized pseudorandom noise and a piezoresistive pressure transducer (Cosmed Quark i2M) while in a seated position, donning a noseclip, and providing manual cheek support in accordance with recommendations (Ref). A grading strategy was implemented to ensure data quality, prioritizing a coefficient of variation ≤ 10% between at least 3 trials. Predicted values for oscillometry parameters (R5, R20, X5, AX) were calculated using published reference equations [18].

## Flow- and nitroglycerin-mediated vasodilation (FMD, NMD)

FMD of the brachial artery was assessed via Doppler ultrasound (GE Vivid E90) using a 12-MHz linear array transducer with a custom-designed support in accordance with current guidelines [19]. In brief, images and velocities were obtained

in duplex mode, corrected with an insonation angle of 60° with the range gate (~1.5 mm) set in the mid-vessel. Once positioned, there was a continuous recording for 3 min after which a blood pressure cuff was inflated to 220–250 mmHg for 5 min and then deflated to induce reactive hyperemia and monitored for a 3 min recovery period. Offline analyses of brachial artery diameters and velocities were performed using automated, edge-detection software (Cardiovascular Suite®, FMD Studio; Pisa, Italy). Variables of interest included endothelium-dependent FMD and endothelium-independent (nitroglycerine administration) NMD percentage change (% change = (peak diameter − baseline diameter)/baseline diameter *100).

## Flow cytometric analysis

Detailed gating strategies and selection of monocyte markers are outlined in Figs 1 and 2. Briefly, monocytes were classified into subsets based on paradigms which define three phenotypes (classical, intermediate, and non-classical) distinguished by CD14 and CD16 expression. The progression from classical to non-classical monocytes within the circulation reflects their activation status and functional specialization over time (Fig 1A) [11,12]. Monocyte markers used for this

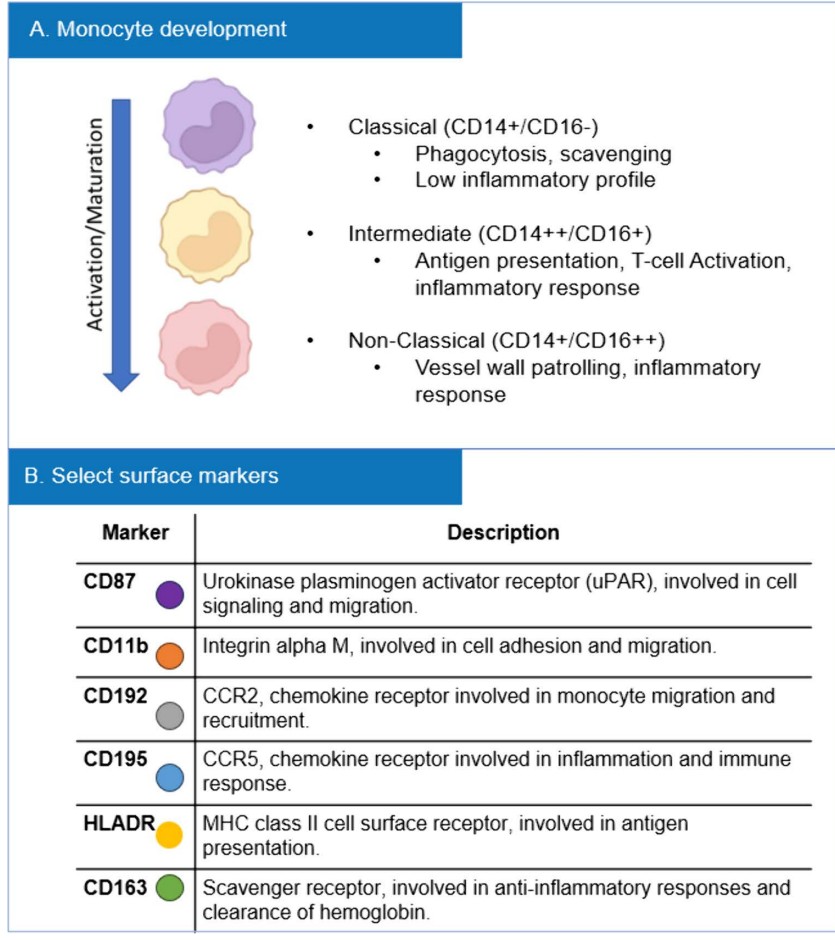

**Fig 1. Monocyte subset maturation and marker selection. (A)** After exiting the bone marrow and entering circulation, monocytes transition from classical to intermediate and then non-classical subsets in a linear fashion, dependent on time and inflammatory activation signals. **(B)** Selected monocyte markers and corresponding functions.

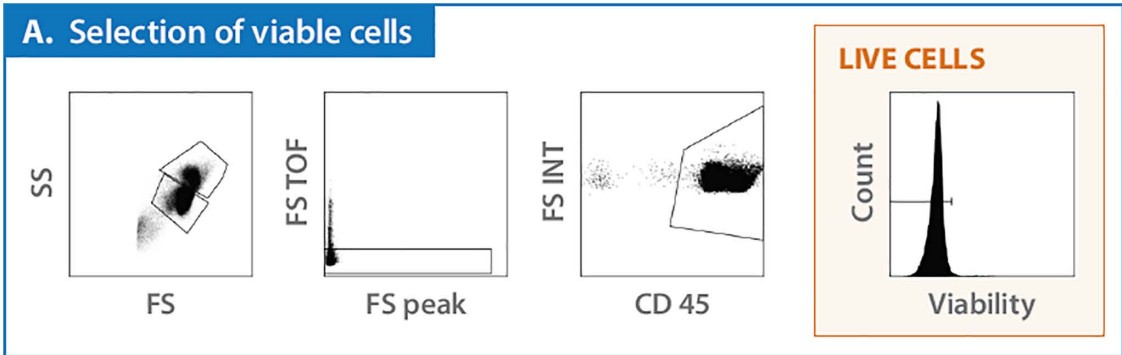

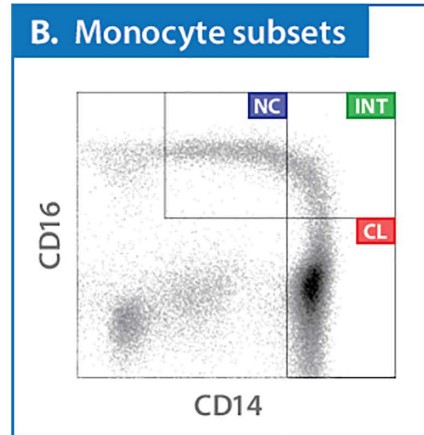

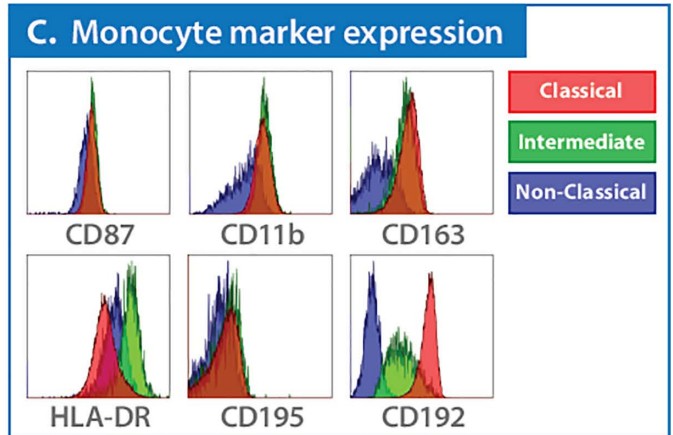

**Fig 2. Flow cytometric gating strategy.** Peripheral blood samples, collected at screening, were processed by magnetic separation to isolate subpopulations. **(A)** Cells were divided into CD15+ (polymorphonuclear neutrophils; PMN) and CD15- (peripheral blood mononuclear cells; PBMCs) fractions. The PBMC fraction was further analyzed by flow cytometry, initially gated on forward and side scatter properties to identify monocyte and lymphocyte populations. Gating on the common leukocyte antigen CD45 was employed to refine these populations and exclude red blood cells, and cells were stained with a viability dye to ensure analysis of live cells only. **(B)** From the viable cell population, monocytes were categorized into three subsets based on their expression of CD14 and CD16. **(C)** Within each monocyte subset, further analysis focused on six specific markers—CD87, CD11b, CD163, HLA-DR, CD195, and CD192. Mean fluorescence intensity (MFI), measured in arbitrary units (a.u.), quantitatively reflects the density of marker expression on the cell surface, providing a more detailed assessment of cellular activation states to evaluate the level of expression rather than just the presence (frequency) of these markers. Further gating to evaluate a broad lymphocyte population was conducted and is included in the supplemental material.

study included CD87, CD11b, CD192, CD195, HLA-DR, and CD163, which were selected based on their distinct roles in monocyte function (Fig 1B).

## Data preparation and principal component analysis

The dataset comprised expression levels of six CD markers (CD87, CD11b, CD192, CD195, HLA-DR, and CD163) across three monocyte subsets (classical, intermediate, and non-classical), resulting in a total of 18 parameters per subject. To reduce dimensionality and capture the most variance in the data, Principal Component Analysis (PCA) was performed on the scaled marker expression data to ensure that each marker contributed equally to the analysis. The first 6 principal components (PCs) were selected for use in the hierarchical clustering analysis. This choice was guided by several considerations: i) variance explained – The first 3 PCs accounted for over 80% of the variance, a common threshold for dimensionality reduction. Extending to the first 6 PCs captured an additional 15% variance, totaling 95%, thereby including more

subtle but potentially important differences in the data, ii) cluster performance – hierarchical clustering on the first 6 PCs yielded 2 distinct clusters and 4 subclusters with relatively balanced groups, allowing a more granular understanding of the data compared to clustering with fewer PCs, iii) cluster interpretability – the clusters identified using the first six PCs aligned with known biological characteristics and allowing for a more targeted analysis.

Hierarchical clustering was performed on the selected PCs using the Euclidean distance metric and Ward's method to minimize the total within-cluster variance. The resulting groups were then interpreted by examining the frequencies of each CD marker within each cluster, providing insights into the biological significance of the clusters, particularly in terms of monocyte subset activation and marker expression profiles.

## Statistical analysis

All statistical analyses were conducted using R version 3.5.1 (R Foundation for Statistical Computing, Vienna, Austria) [20]. Continuous variables were reported as either means and standard deviations (mean±SD) or medians and inter-quartile ranges (median [IQR]), as appropriate. Normality of data distributions was assessed using a combination of the Shapiro-Wilk test and visual inspection, particularly for smaller sample sizes.

Categorical variables were compared using either the chi-square test with continuity correction or Fisher's exact test, depending on the sample size and expected cell counts. Group comparisons for continuous variables (including CD marker expression, PFT, FOT, and FMD measures) were performed using linear models, with monocyte cluster as the primary predictor and smoking status (never/ever) as a covariate. Estimated marginal means and their standard errors were calculated and pairwise comparisons between the adjusted means of cluster 1 and cluster 2 were performed. Effect sizes were derived from the linear regression models adjusted for smoking history. Effect sizes were calculated for each parameter as: the difference in estimated marginal means between groups divided by the residual standard deviation of the respective model to yield Cohen's d. Bonferroni correction was applied for multiple comparisons [21]. Where appropriate, data were log-transformed to meet the assumptions of parametric testing, or Kruskal-Wallis tests were applied for non-parametric data. A p-value of less than 0.05 was considered statistically significant for all analyses.

## Results

### Subject characteristics

Out of the 84 subjects enrolled, 2 subjects were excluded due to poor data quality, resulting in a final cohort of 82 subjects (Fig 3). Six subjects were excluded from FOT analysis – four due to missing data and two due to poor data quality. A smaller portion (n=48) of the total sample completed the FMD and NMD procedures. The mean age was 41.3 years (± 8.68), and the mean body mass index (BMI) was 29.1 kg/m² (± 3.91). Most participants identified as male (88.46%) and White (81.71%). Of the participants, 20.73% reported ever smoking history. The sample reported low daily symptom burden, but nearly half experienced difficulty with mild-moderate physical activity. Additionally, most subjects reported exposure to multiple environmental hazards during deployment (Table 1).

### Monocyte subsets display heterogenous marker expression

The mean frequencies for the classical (CL), intermediate (INT), and non-classical (NC) monocyte subsets were 85.7% (± 6.22%), 4.7% (± 2.01%), and 9.6% (± 4.94%) respectively. Analysis revealed significant differences between each subset for measured markers CD11b, CD163, CD192, CD195, CD87, and HLA-DR (Fig 4). CL monocytes had the highest expression of CD192, while CD87, CD11b, and CD163 were notably higher in both CL and INT subsets compared to NC (p<0.001). INT monocytes exhibited the highest levels of HLA-DR (p<0.001), while NC monocytes display the highest expression of CD195 (p<0.001).

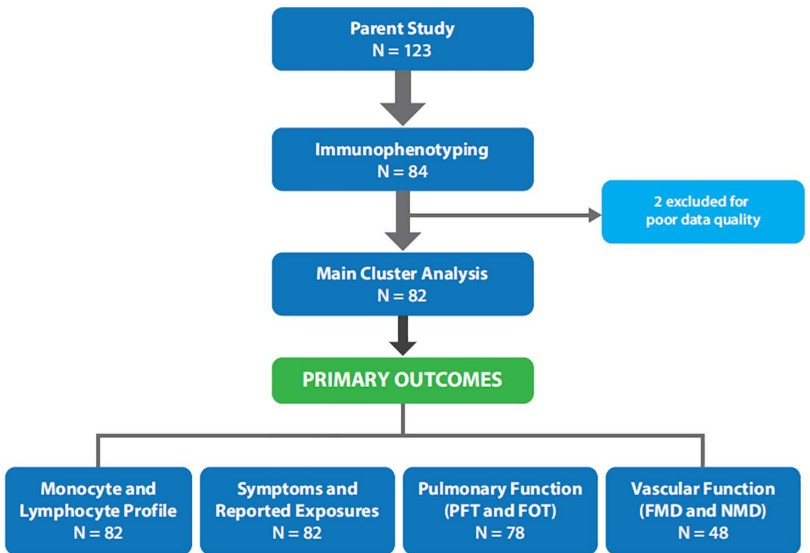

**Fig 3. Flow chart for final dataset.** Eighty-four subjects provided a blood sample for cell isolation and immunophenotyping. Two subjects were excluded for poor data quality. Analysis was completed on primary outcomes of interest including self-reported symptoms, medical history, deployment information as well as physiological measures including PFT, FOT, and FMD/NMD. Only a smaller portion (n = 48) of the total sample completed the FMD and NMD procedures.

## Cluster analysis and patient demographics by cluster

PCA followed by hierarchical clustering identified two main clusters of monocytes that were phenotypically distinct. Cluster 1 was characterized by greater expression of CD87, CD11b, CD192, and CD163 within the CL and INT subsets. Cluster 2 contained a greater percentage of NC monocytes which expressed CD195 (Table 2). There were no significant differences in demographic measures, smoking history (including pack years), or self-reported exposures between clusters, except for total years since first deployment (Table 3). Clusters did not differ in self-reported respiratory symptoms or functional limitations (Table 4). Each main cluster consisted of two subclusters (Fig 5). Our primary analysis for this study were conducted using only the two larger clusters, as the subclusters contained significantly smaller number of cells, reducing the power of our analysis. These subclusters are briefly described later.

## Broad lymphocyte populations analysis

In addition to the monocyte profile presented in the main paper, we briefly examined the lymphocyte population in this sample, broken down into broad B and T-cell populations (S1 Fig). The mean frequencies for the T-helper cells (CD3 + CD4+), cytotoxic T cells (CD3 + CD8+), activated B cells (CD19 + CD74+) were 59.1% (± 9.92), 33.3% (± 9.14) 10.0% (± 4.50), respectively. The yields for activated T-helper cells (CD3 + CD4 + CD74+) and cytotoxic T cells (CD3 + CD8 + CD74+) were extremely low, at 0.1% (± 0.11) and 0.1% (± 0.13) and were excluded from further analysis due to their minimal representation.

Correlation analysis between monocyte subsets and lymphocyte subsets revealed several significant associations. CL monocytes were positively correlated with activated B cells (r = 0.19, p = 0.09). NC monocytes were negatively correlated with activated B cells (r = −0.22, p = 0.047). T-helper cells showed a positive correlation with activated B cells (r = 0.34, p = 0.002), while cytotoxic T cells demonstrated a negative correlation with activated B cells (r = −0.32, p = 0.003).

**Table 1. Study sample characteristics.**

| Characteristic | N = 82[1] |
|---|---|
| Age, years | 41.3 (8.68) |
| BMI, kg/m$^2$ | 29.1 (3.91) |
| Male | 71 (86.59%) |
| Race | |
| American Indian/Alaska Native | 1.0 (1.22%) |
| Asian | 1.0 (1.22%) |
| Black or African-American | 8.0 (9.76%) |
| More than one race | 3.0 (3.66%) |
| Unknown/did not report | 2.0 (2.44%) |
| White | 67.0 (81.71%) |
| Ethnicity | |
| Hispanic or Latino | 24.0 (29.27%) |
| Non-Hispanic or Latino | 57.0 (69.51%) |
| Unknown/did not report | 1.0 (1.22%) |
| Ever smoker, n (% yes) | 17.0 (20.73%) |
| Pack years | 5.3 (3.97) |
| Deployments | |
| Cumulative deployment (days) | 515.4 (336.26) |
| Time since initial deployment (years) | 14.3 (3.84) |
| Time since last deployment (years) | 11.5 (4.19) |
| **Symptoms** | |
| Dyspnea | |
| mMRC > 1 | 23.0 (28.05%) |
| Shortness of breath within past 4 weeks | 16.0 (19.51%) |
| Functional Limitations | |
| Difficulty running or jogging 1 mile on a level surface | 42.0 (51.22%) |
| Difficulty walking on a level surface for one mile | 17.0 (20.73%) |
| Difficulty walking a 1/4 of a mile (about 3 city blocks) | 13.0 (15.85%) |
| Difficulty walking up a hill or incline | 40.0 (48.78%) |
| Difficulty walking up 10 steps or climb a flight of stairs | 30.0 (36.59%) |
| SGRQ Component Scores | |
| Symptoms | 23.5 (18.68) |
| Activity | 17.7 (19.85) |
| Impact | 8.5 (12.40) |
| Total | 13.7 (14.22) |
| **Exposures** | |
| Near a burn pit (on the base or close enough to the base for you to see the smoke) | 68.0 (82.93%) |
| Duties associated with the burn pit (e.g., trash burning, hauling trash, providing security at burn pit, trash sorting) | 34.0 (50.00%) |
| Close enough to feel the blast of an IED (improvised explosive device) or other explosive device | 53.0 (64.63%) |
| Near heavy smoke from weapons, markers or other combat items that made breathing difficult | 44.0 (53.66%) |
| In a convoy or other vehicle operations one or more times per week where you felt you breathed in large amounts of dust or vehicle exhaust fumes | 58.0 (70.73%) |
| Perform refueling operations on a daily basis | 29.0 (35.37%) |
| Experience dust storms during deployment | 69.0 (84.15%) |

[1] Mean (SD); n (%).

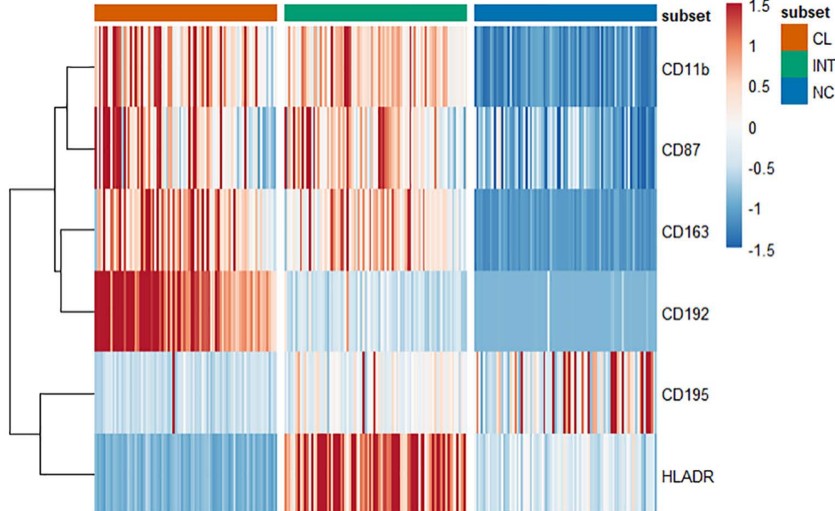

**Fig 4. Individual marker expression within each monocyte subset.** Heatmap showing individual marker expression (MFI) within each subset, with every column representing a single subject's expression (repeated for each subset) and every row an individual marker. Red indicates higher standardize frequency (z-score), and blue lower.

Additionally, there were no significant differences between our main clusters, cluster 1 and cluster 2, in T-helper (59.2% ± 9.9%, 58.9% ± 10.0%; p > 0.9), cytotoxic T cell (33.5% ± 8.7%, 33.0% ± 10.0%; p = 0.8), and activated B cell (10.0% ± 4.1%, 10.1% ± 5.1%; p = 0.8) populations.

### Pulmonary and vascular function within clusters

As expected, based on the inclusion criteria of the parent study, spirometry values and lung volumes and capacities fell within normal predicted ranges [17]. Additionally, there was no reduction in $DL_{CO}$. There were no significant differences between measures of pulmonary function ($FEV_1$, FVC, and the $FEV_1$/FVC ratio), diffusing capacity ($DL_{CO}$), or fractional exhaled nitric oxide (FeNO) between clusters (Table 5).

Despite no differences in standard PFT measures Cluster 2 exhibited significantly greater resistance at both R5 (p = 0.017) and R20 (p = 0.045), indicating more generalized airway obstruction (Table 6). The % predicted AX was higher in cluster 2, although this did not reach statistical significance (p = 0.071). Endothelial-dependent vasodilation, as measured by the FMD/NMD ratio, was significantly lower in cluster 2, suggesting impaired endothelial function (Table 7). Due to the smaller sample size of those with available FMD/NDM data (N = 48), we compared key demographic characteristics (e.g., age, BMI, sex, smoking status) and self-reported deployment factors between participants who completed FMD/NMD testing and those who did not. Our comparison revealed no statistically significant differences in age (p = 0.4), BMI (p = 0.9), sex (p = 0.9), race (p = 0.4), smoking status (p = 0.8), pack years (p = 0.6), mMRC (p = 0.3), total deployment time and time since both first and last deployments (all P > 0.3) between those who underwent FMD/NMD testing and those who did not.

### Discussion

The present study sought to characterize monocyte phenotypes and functions, and their association with physiological outcomes, among previously deployed veterans with airborne hazards exposure and respiratory symptoms. Using a non-biased clustering analysis, we were able to identify two different clusters on the basis of their expression of monocyte markers. Veterans assigned to these two distinct clusters also exhibited significant differences

**Table 2. Cluster differences in CD surface marker expression.**

| Characteristic | Overall, N = 82[1] | Cluster 1 N = 51[1] | Cluster 2 N = 31[1] | p-value[2] | Effect Size[3] |
|---|---|---|---|---|---|
| **Subset Frequencies (%)** | | | | | |
| Classical | 86.9 (81.60, 90.0) | 87.3 (82.38, 91.3) | 86.6 (81.06, 88.4) | 0.2 | 0.26 |
| Intermediate | 4.6 (3.04, 6.1) | 4.6 (2.84, 6.2) | 4.2 (3.18, 6.1) | 0.8 | 0.04 |
| Non-Classical | 8.4 (6.03, 11.9) | 7.9 (5.89, 10.7) | 9.1 (7.17, 12.6) | 0.069 | −0.33 |
| **CD87** | | | | | |
| Classical | 6.4 (5.41, 7.8) | 7.1 (6.20, 8.9) | 5.3 (4.54, 6.0) | **<0.001** | 1.3 |
| Intermediate | 6.7 (5.80, 7.6) | 7.1 (6.35, 7.8) | 5.8 (5.08, 6.4) | **<0.001** | 1.2 |
| Non-Classical | 4.2 (3.60, 4.9) | 4.4 (3.96, 5.1) | 3.7 (2.96, 4.2) | **<0.001** | 0.94 |
| **CD11b** | | | | | |
| Classical | 14.3 (11.65, 17.9) | 15.0 (13.38, 19.3) | 11.7 (10.71, 14.0) | **<0.001** | 0.85 |
| Intermediate | 14.3 (12.27, 15.8) | 15.0 (13.28, 16.1) | 12.7 (11.80, 14.6) | **0.005** | 0.69 |
| Non-Classical | 4.4 (3.46, 5.2) | 4.7 (3.47, 5.3) | 3.9 (3.44, 4.5) | 0.075 | 0.40 |
| **CD192** | | | | | |
| Classical | 19.3 (15.74, 23.0) | 21.6 (17.08, 25.6) | 16.8 (14.44, 18.6) | **<0.001** | 1.2 |
| Intermediate | 4.4 (3.21, 5.6) | 4.5 (3.62, 6.1) | 3.8 (2.91, 4.7) | 0.071 | 0.44 |
| Non-Classical | 0.8 (0.75, 0.8) | 0.8 (0.73, 0.8) | 0.8 (0.76, 0.9) | **0.002** | 0.21 |
| **CD195** | | | | | |
| Classical | 3.8 (3.69, 3.9) | 3.8 (3.68, 3.9) | 3.8 (3.73, 3.9) | 0.13 | −0.34 |
| Intermediate | 4.3 (4.02, 4.5) | 4.1 (3.96, 4.4) | 4.4 (4.24, 4.6) | **<0.001** | −0.74 |
| Non-Classical | 4.4 (3.84, 5.1) | 3.9 (3.72, 4.4) | 5.4 (4.93, 6.4) | **<0.001** | −1.3 |
| **HLA-DR** | | | | | |
| Classical | 11.3 (8.49, 13.6) | 11.5 (8.83, 13.9) | 10.6 (8.25, 12.3) | 0.2 | 0.24 |
| Intermediate | 79.7 (65.25, 101.2) | 79.6 (64.95, 101.8) | 85.2 (67.87, 100.5) | 0.8 | −0.04 |
| Non-Classical | 28.9 (22.08, 37.4) | 26.1 (21.72, 37.6) | 29.1 (23.11, 37.1) | >0.9 | −0.01 |
| **CD163** | | | | | |
| Classical | 4.8 (3.86, 6.0) | 5.0 (4.30, 6.2) | 4.2 (3.29, 5.3) | **0.007** | 0.58 |
| Intermediate | 4.3 (3.52, 5.1) | 4.5 (3.67, 5.5) | 4.1 (3.28, 4.7) | 0.072 | 0.47 |
| Non-Classical | 1.2 (1.05, 1.3) | 1.2 (1.06, 1.3) | 1.1 (1.04, 1.2) | 0.2 | 0.33 |

[1] Median (IQR); [2] Wilcoxon rank sum test; [3] Cohen's d.

in airway impedance and vascular function. Cluster 1 is characterized by elevated markers associated with an effective immune response with enhanced capacity for recruitment (CD192), tissue adhesion and infiltration (CD87, CD11b), and anti-inflammatory responses (CD163) [22–24]. When viewed overall, this phenotype has the capacity to participate in the resolution of inflammatory signaling, and to mitigate inflammation and promote healing. In contrast, cluster 2, characterized by its higher expression of CD195 and greater frequency of non-classical monocytes is more typical of a chronic inflammatory state with the capacity for prolonged monocyte recruitment and retention at inflammatory sites and sustained lung and vascular dysfunction. Importantly, differences in CD192 and CD195 expression by these two clusters appears to reflect their functional roles. CD192 (CCR2) primarily responds to CCL2 (MCP-1), a potent chemoattractant, crucial for initial recruitment of monocytes. CD195 (CCR5) responds to a broader range of chemokines (CCL3, CCL4, CCL5) which can fine-tune recruitment and activation of both monocytes and other immune cells.

**Table 3. Subject demographics and self-reported deployment history by cluster.**

| Demographics | Overall N = 82[1] | Cluster 1 N = 51[1] | Cluster 2 N = 31[1] | p-value[2] |
|---|---|---|---|---|
| Age, years | 41.3 (8.68) | 41.1 (9.15) | 41.6 (7.99) | 0.7 |
| BMI, kg/m$^2$ | 29.1 (3.91) | 29.4 (3.89) | 28.4 (3.93) | 0.15 |
| Male, n (%) | 71.0 (86.59%) | 44.0 (86.27%) | 27.0 (87.10%) | >0.9 |
| Race | | | | 0.6 |
| American Indian/Alaska Native | 1.0 (1.22%) | 0.0 (0.00%) | 1.0 (3.23%) | |
| Asian | 1.0 (1.22%) | 1.0 (1.96%) | 0.0 (0.00%) | |
| Black or African-American | 8.0 (9.76%) | 6.0 (11.76%) | 2.0 (6.45%) | |
| More than one race | 3.0 (3.66%) | 1.0 (1.96%) | 2.0 (6.45%) | |
| Unknown/did not report | 2.0 (2.44%) | 1.0 (1.96%) | 1.0 (3.23%) | |
| White | 67.0 (81.71%) | 42.0 (82.35%) | 25.0 (80.65%) | |
| Ethnicity | | | | 0.4 |
| Hispanic or Latino | 24.0 (29.27%) | 14.0 (27.45%) | 10.0 (32.26%) | |
| Non-Hispanic or Latino | 57.0 (69.51%) | 37.0 (72.55%) | 20.0 (64.52%) | |
| Unknown/did not report | 1.0 (1.22%) | 0.0 (0.00%) | 1.0 (3.23%) | |
| Ever smoker, n (% yes) | 17.0 (20.73%) | 10.0 (19.61%) | 7.0 (22.58%) | 0.7 |
| Pack years | 5.3 (3.97) | 5.2 (3.93) | 5.5 (4.4) | 0.8 |
| Dyspnea (mMRC) | 23.0 (28.05%) | 14.0 (27.45%) | 9.0 (29.03%) | 0.9 |
| Deployment | | | | |
| Cumulative deployment (days) | 515.4 (336.26) | 513.0 (354.30) | 519.4 (309.92) | 0.5 |
| Time since initial deployment (years) | 14.3 (3.84) | 13.7 (4.05) | 15.3 (3.29) | **0.039** |
| Time since last deployment (years) | 11.5 (4.19) | 11.1 (4.31) | 12.3 (3.91) | 0.14 |

[1]Mean (SD); n (%); [2] Wilcoxon rank sum test; Wilcoxon rank sum exact test; Fisher's exact test; Pearson's Chi-squared test.

### Differing roles of intermediate and non-classical monocytes

Unlike prior studies linking increased INT monocytes with inflammatory diseases [25], we did not observe an association between INT monocytes and pulmonary or vascular function in this cohort. In contrast, cluster2 – with greater NC monocytes, lower CD192 and elevated CD195 expression – was associated with greater airway resistance and peripheral vascular dysfunction. This finding suggests that NC monocytes act to sustain chronic inflammation by facilitating ongoing monocyte recruitment and activation at inflammatory sites [26,27]. While this is one possible explanation for the increased dysfunction identified in cluster 2, further research is needed to clarify the specific mechanism by which NC monocytes and varying expression of surface markers may contribute to vascular impairments in these veterans. Overall, these findings highlight the complexity of monocyte behavior and the need for more nuanced understanding of how these cells function in exposure related disease.

### Monocyte profiles and functional implications

Consistent with prior literature, pulmonary function appears preserved among our subjects and similar between clusters [28]. Respiratory oscillometry, however, differed such that total airway resistance (R5) was elevated among those in cluster 2 relative to cluster 1. Similar moderate effect sizes, albeit not statistically significant, were also observed for central airway resistance (R20) and reactance area (AX). Relative to cluster 1, cluster 2 exhibited an oscillometry pattern characterized by reduced airway caliber and increased stiffness. These features are suggestive of airway remodeling secondary to deployment exposures brought about by chronic inflammation [29,30]. This interpretation lacks direct evidence but

**Table 4. Self-reported symptoms, functional limitations, and exposures cluster.**

| Symptoms, Limitations, and Exposures | Overall N=82[1] | Cluster 1 N=51[1] | Cluster 2 N=31[1] | p-value[2] |
|---|---|---|---|---|
| **Symptoms** | | | | |
| Current cough for more than 3 weeks | 7.0 (8.54%) | 4.0 (7.84%) | 3.0 (9.68%) | >0.9 |
| Current sputum or phlegm production | 19.0 (23.17%) | 12.0 (23.53%) | 7.0 (22.58%) | >0.9 |
| Current wheezing or whistling in the chest | 7.0 (8.54%) | 3.0 (5.88%) | 4.0 (12.90%) | 0.4 |
| Current decreased ability to exercise | 24.0 (29.27%) | 14.0 (27.45%) | 10.0 (32.26%) | 0.6 |
| Current chronic sinus infection/sinusitis | 14.0 (17.07%) | 9.0 (17.65%) | 5.0 (16.13%) | 0.9 |
| **Functional Limitations** | | | | |
| Difficulty running or jogging 1 mile on a level surface | 42.0 (51.22%) | 24.0 (47.06%) | 18.0 (58.06%) | 0.3 |
| Difficulty walking on a level surface for one mile | 17.0 (20.73%) | 14.0 (27.45%) | 3.0 (9.68%) | 0.054 |
| Difficulty walking a 1/4 of a mile (about 3 city blocks) | 13.0 (15.85%) | 10.0 (19.61%) | 3.0 (9.68%) | 0.4 |
| Difficulty walking up a hill or incline | 40.0 (48.74%) | 26.0 (50.98%) | 14.0 (45.16%) | 0.6 |
| Difficulty walking up 10 steps or climb a flight of stairs | 30.0 (36.59%) | 19.0 (37.25%) | 11.0 (35.48%) | 0.9 |
| **Exposures** | | | | |
| Near a burn pit, close enough to see smoke | 68.0 (82.93%) | 42.0 (82.35%) | 26.0 (83.87%) | 0.9 |
| Duties included the burn pit | 34.0 (50.00%) | 21.0 (50.00%) | 13.0 (50.00%) | >0.9 |
| Unknown | 14 | 9 | 5 | |
| Close enough to feel the blast of an IED (improvised explosive device) or other explosive device | 53.0 (64.63%) | 31.0 (60.78%) | 22.0 (70.97%) | 0.3 |
| Near heavy smoke from weapons, markers or other combat items that made breathing difficult | 4.0 (53.66%) | 28.0 (54.90%) | 16.0 (51.61%) | 0.8 |
| In a convoy or other vehicle operations one or more times per week where you felt you breathed in large amounts of dust or vehicle exhaust fumes | 58.0 (70.73%) | 36.0 (70.59%) | 22.0 (70.97%) | >0.9 |
| Performed refueling operations on a daily basis | 29.0 (35.37%) | 17.0 (33.33%) | 12.0 (38.71%) | 0.6 |
| Experienced dust storms during deployment | 69.0 (84.15%) | 42.0 (82.35%) | 27.0 (87.10%) | 0.8 |

[1] n (%); [2] Fisher's exact test; Pearson's Chi-squared test.

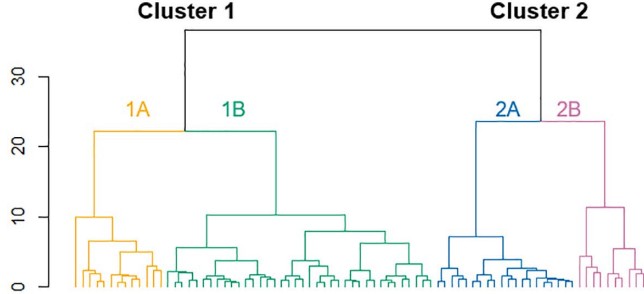

**Fig 5. Dendrogram of monocyte cluster analysis based on PCA.** Hierarchical clustering of monocyte surface markers following dimensionality reductions with PCA. Two major clusters emerged with distinct CD expression phenotypes potentially reflective of differing activation states. Four additional subclusters were identified and discussed within the context of monocyte activation states.

aligns with the known roles of non-classical monocytes in maintaining prolonged inflammatory responses [27]. Of note, our findings that fractional exhaled nitric oxide (FeNO) was similar between groups indicates that airway inflammation present among cluster 2 is not primarily driven by eosinophils. Non-eosinophilic or neutrophilic type airway inflammation is more often associated with occupational exposure and less responsive to conventional therapies [31]. The present study

**Table 5. Pulmonary function by cluster.**

| PFTs | Overall N = 79[1] | Cluster 1 N = 51[1] | Cluster 2 N = 31[1] | p-value[2] | Effect Size[3] |
|---|---|---|---|---|---|
| FVC pp | 106.6 (13.66) | 106.8 (13.36) | 106.4 (14.42) | >0.9 | 0.03 |
| $FEV_1$ pp | 105.9 (13.76) | 106.1 (13.48) | 105.4 (14.45) | 0.828 | 0.05 |
| % $FEV_1$/FVC | 80.3 (5.65) | 80.5 (5.43) | 80.0 (6.10) | 0.738 | 0.08 |
| $DL_{CO}$ pp | 98.4 (12.78) | 97.7 (12.06) | 99.6 (14.08) | 0.564 | −0.13 |
| RV pp | 76.6 (32.19) | 73.3 (35.05) | 82.1 (26.35) | 0.242 | −0.28 |
| TLC pp | 95.2 (11.77) | 94.7 (11.15) | 96.0 (12.91) | 0.638 | −0.11 |
| % RV/TLC | 19.3 (7.97) | 18.6 (9.06) | 20.4 (5.63) | 0.313 | −0.24 |
| FeNO | 25.1 (16.25) | 24.9 (18.69) | 25.3 (14.18) | >0.9 | −0.02 |

[1]Unadjusted group means ± standard deviation. [2] Comparison between the adjusted means of cluster 1 and cluster 2, derived from linear regression models adjusted for smoking history. [3] Cohen's d.

**Table 6. Lung function assessed by oscillometry.**

| Oscillometry | Overall, N = 76[1] | 1, N = 47[1] | 2, N = 29[1] | p-value[2] | Effect Size[3] |
|---|---|---|---|---|---|
| R5 pp | 131.1 (42.16) | 121.9 (36.41) | 146.0 (47.02) | **0.017** | −0.58 |
| R20 pp | 132.4 (42.40) | 124.5 (37.01) | 145.2 (47.87) | 0.045 | −0.48 |
| % R5-R20/R5 | 6.1 (12.35) | 6.0 (13.79) | 6.4 (9.81) | 0.887 | 0.03 |
| X5 pp | 128.4 (40.35) | 122.8 (39.06) | 137.5 (41.41) | 0.113 | −0.38 |
| AX pp | 256.6 (174.48) | 228.3 (157.58) | 302.4 (192.97) | 0.071 | −0.43 |

[1]Unadjusted group means ± standard deviation. [2] Comparison between the adjusted means of cluster 1 and cluster 2, derived from linear regression models adjusted for smoking history. [3] Cohen's d.

did not include sputum samples to support this interpretation; therefore, future studies should consider both airway and circulating markers of inflammation.

Veterans comprising cluster 2 also demonstrated impaired peripheral vascular function, as indicated by a lower percentage FMD and a reduced FMD/NMD ratio. The preservation of NMD suggests that vascular smooth muscle responsiveness remains intact, while the impaired FMD reflects an endothelial-specific dysfunction, potentially due to chronic NO scavenging driven by inflammation. Overall, this may reflect a diminished capacity of the endothelium to either generate or utilize NO effectively [32]. This is particularly relevant given the inflammatory profile of cluster 2, where sustained inflammation may impair endothelial health and function via scavenging of endothelial derived NO, potentially leading to long-term vascular or metabolic complications [32,33].

Despite these differences in endothelial function and airway oscillometry, we did not observe any significant differences in diffusing capacity of the lung for carbon monoxide ($DL_{CO}$) between the clusters. $DL_{CO}$ is often used as an indicator of alveolar-capillary membrane integrity and gas exchange efficiency, and we initially hypothesized that the inflammatory profiles, like the one observed in cluster 2, would result in impaired gas exchange [34]. However, the absence of differences in $DL_{CO}$ between the clusters suggests that while chronic inflammation in cluster 2 is potentially affecting small airways and endothelial (peripheral vascular) function, it has not yet progressed to the point of significantly altering the alveolar-capillary interface. This observation also suggests a more subtle form of endothelial dysfunction, one that affects the microvasculature without causing large-scale disruption in gas exchange. Overall, our findings provide a preliminary link between monocyte profiles and vascular health, but additional studies are necessary to fully understand how inflammatory dysregulation contributes to vascular dysfunction.

**Table 7. Flow mediated vasodilation by cluster.**

| Characteristic | Overall N = 48[1] | Cluster 1 N = 31[1] | Cluster 2 N = 17[1] | p-value[2] | Effect Size[3] |
|---|---|---|---|---|---|
| FMD % | 8.0 (3.56) | 8.8 (3.72) | 6.7 (2.90) | **0.058** | 0.59 |
| NMD % | 16.9 (4.02) | 15.9 (3.46) | 18.8 (4.49) | **0.023** | −0.75 |
| FMD/NMD | 0.5 (0.22) | 0.6 (0.22) | 0.4 (0.20) | **0.019** | 0.77 |

[1]Unadjusted group means ± standard deviation. [2] Comparison between the adjusted means of cluster 1 and cluster 2, derived from linear regression models adjusted for smoking history. [3] Cohen's d.

## Exploratory subclusters

In addition to the two primary clusters, we identified two subclusters within each main cluster: subclusters 1A and 1B within cluster 1, and subclusters 2A and 2B within cluster 2. The purpose of this subcluster analysis is to highlight the heterogeneity within each of the main clusters. Due to smaller sample size, this sub-cluster analysis was exploratory and statistical comparisons should be interpreted cautiously. However, qualitative assessment reveals distinct immunophenotypic profiles that may have important biological and clinical implications for this population. These profiles are presented in Fig 6, and the full table showing marker differences between subclusters can be found in the supplemental material.

The subcluster analysis revealed that within both cluster 1 and cluster 2, there are distinct phenotypes that further differentiate monocyte functionality. Within cluster 1, subcluster 1A is characterized by heightened migratory and adhesive markers (CD87, CD11b) and reduced antigen-presenting capabilities (HLA-DR) highlighting a potentially 'primed' classical phenotype. In contrast, subcluster 1B shows moderate expression of these markers but is distinguished by high HLA-DR expression – balancing rapid response with adaptive immune interactions. Within cluster 2, subcluster 2A exhibits reduced migratory (CD87, CD11b, CD192) markers compared to all other subclusters, while retaining moderate CD195 expression (second only to subcluster 2B). However, subcluster 2A is mostly defined by the largest expansion of the non-classical subset overall. In summary, the combination of low migratory markers with high NC monocytes suggests a phenotype that aligns with the previously described regulatory or vessel patrolling role [35]. Finally, subcluster 2B exhibited the highest CD195 and HLA-DR expression, alongside elevated CD163, suggesting an over-activated monocyte phenotype likely associated with chronic inflammation. Interestingly, subcluster D also showed the greatest percentage predicted AX values, consistent with poorer lung compliance and airway resistance. However, this pattern was not mirrored in endothelial function measures (FMD).

Due to the reduced sample sizes within the subclusters, these observations should be interpreted with cautious. Additionally, there are notable overlaps and similarities among the clusters that merit attention when interpreting these groups. For instance, subclusters 2A and 2B were mostly grouped together due to their lower CD192 and elevated CD195 expression, which were heavily weighted by the first three components in our PCA analysis. CD195 is a key marker for chemotaxis and interaction with endothelial cells and its high expression in these subclusters underscores their regulatory and endothelial-related functions. Additionally, while we focused on markers such as CD87, CD11b, and CD192 which define a more primed or acute inflammatory phenotype in cluster 1 we recognize that these markers are not exclusive to the roles described here and play parts of both inflammatory and anti-inflammatory pathways. While these findings provide important insights, more robust studies are needed to further validate the role of monocyte subsets in both lung function and inflammation.

## Limitations

A key limitation of our study is the absence of a control group of non-deployed veterans or individuals without unexplained respiratory symptoms. Without these baseline data, our findings are relative to internal variations between clusters within our study population, rather than deviations from a healthy or unaffected group. As such, the results provide insight into the

| Subcluster 1A: Activated Classical | Subcluster 1B: Moderately Activated |
|---|---|
| **Key Features:**<br>↑ Largest classical population<br>↑ High CD87, CD11b, CD192<br>↓ Low HLA-DR<br><br>**Summary:**<br>A potentially 'primed' classical phenotype with heightened migratory and adhesive markers (CD87, CD11b) and reduced antigen-presenting capabilities (HLA-DR). | **Key Features:**<br>↔Moderate CD87, CD11b CD192<br>↑ High HLA-DR<br><br>**Summary:**<br>A more balanced phenotype, potentially maintaining capabilities for a rapid response with adaptive immune interactions |
| Subcluster 2A: High Non-classical | Subcluster 2B: Uniquely Activated |
| **Key Features:**<br>↑ Largest non- classical population<br>↑ Moderate CD195<br>↓ Lowest CD87, CD11b, CD192<br><br>**Summary:**<br>A combination of the highest in non-classical monocytes with lower migratory and adhesive markers (CD87, CD11b, CD192) indicative of the previously describe vessel patrolling phenotype. | **Key Features:**<br>↑ Highest CD195 and HLA-DR<br>↑ Moderate CD163<br><br>**Summary:**<br>A unique phenotype with the highest expression of CD195 and HLA-DR, aligning with an overactive phenotype that is unable to resolve inflammation. |

**Fig 6. Summary of subcluster analysis.**

differences between clusters rather than absolute measures of abnormality. Another limitation involves PCA and clustering methods we employed. While these techniques are powerful tools for uncovering patterns in complex data, they are sensitive to the input variables and assumptions used. The clusters identified represent one possible interpretation of the data and should be considered exploratory. Finally, our study provides only a single snapshot in time, with blood samples collected long after potential deployment-related exposures. While this timing is valuable for assessing persistent, long-term immune alterations, it also raises questions as to whether the observed monocyte phenotypes reflect a stable, 'memorized' immune deviation or a more dynamically ongoing inflammatory process. Given the relatively short lifespan of circulating monocytes, their surface marker profiles likely indicate a current or recent activation state. However, such states could be sustained by long-term factors, including potential epigenetic reprogramming of myeloid progenitor cells- a concept akin to innate immune memory or training -or by persistent low-grade inflammation secondary to deployment-related exposures. Our cross-sectional design cannot definitively distinguish between these possibilities, limiting our understanding of how these monocyte profiles evolve over time or in relation to different stages of exposure or disease progression.

## Conclusion and future directions

The current study provides new insights into the monocyte phenotypic profiles potentially driving chronic inflammation in veterans with respiratory symptoms linked to airborne hazard exposures. While we propose that non-classical monocytes

and the upregulation of CD195 may play a role in mediating long-term lung and vascular dysfunction, this hypothesis requires validation through further research. The evolving understanding of monocyte behavior underscores the need for more detailed mechanistic studies to unravel the complex interactions between immune cells and tissue dysfunction.

Future research, including longitudinal studies, would be necessary to track the dynamics of these immune phenotypes and their long-term impact on health. Such studies would be significantly enhanced by serial immune profiling at multiple time points post-deployment. Furthermore, the incorporation of additional biomarkers could provide a more comprehensive picture; this might include plasma measures of systemic inflammation (e.g., cytokines, chemokines), markers of endothelial activation or injury, and potentially analyses of epigenetic modifications within monocyte populations themselves. This multi-faceted approach could help to better elucidate the mechanisms driving the persistence of these monocyte alterations and distinguish between stable long-term changes and more active, ongoing inflammatory signaling in veterans with deployment-related respiratory symptoms.

Despite the significant burden of chronic symptoms in this population—affecting an estimated 4.7 million deployed veterans—there have been no treatment trials to date aimed at addressing the underlying mechanisms of disease. Our findings point to the potential for monocyte-targeted therapies as a novel avenue for intervention, but more research is urgently needed to better understand the inflammatory pathways at play and to identify actionable targets. This study is among the first to examine these immune phenotypes in this population, highlighting the critical need for further work to elucidate the mechanisms driving chronic inflammation and identify strategies for therapeutic intervention.

## Supporting information

**S1 Fig. Extended gating strategy including broad population of B and T-cells.**
(DOCX)

**S1 Table. mMRC (Modified Medical Research Council) Dyspnea Scale.**
(DOCX)

**S2 Table. Selected CD marker overview.**
(DOCX)

**S3 Table. Cell subsets and functions.**
(DOCX)

**S4 Table. Pairwise comparisons of cluster 1 and cluster 2 for each marker, corrected for multiple comparisons.**
(DOCX)

**S5 Table. Subcluster differences in CD surface marker expression.**
(DOCX)

**S6 Table. Pairwise comparisons of subcluster 1A and subcluster 1B for each marker, corrected for multiple comparisons.**
(DOCX)

**S7 Table. Pairwise comparisons of subcluster 2A and subcluster 2B for each marker, corrected for multiple comparisons.**
(DOCX)

**S1 File. R code for statistical analysis.**
(RMD)

## Acknowledgments

The authors thank the Veterans who volunteered for this study and Samy B., Eager N., Domanski H., Phen S., Klein-Adams J., Ndirangu D., Alexander T., Piskura N., Watson M., Therkorn J., Wentz A., Abitante T., Cervelli J., Savoy B., Abramova E.

## Author contributions

**Conceptualization:** Howard Kipen, Anays Sotolongo, Andrew Gow.

**Data curation:** Gregory Pappas, Changjiang Guo.

**Formal analysis:** Gregory Pappas.

**Funding acquisition:** Debra Laskin, Howard Kipen, Michael Falvo, Andrew Gow.

**Investigation:** Gregory Pappas, Carol Gardner, Changjiang Guo, Raymond Rancourt, Andrew Gow.

**Methodology:** Gregory Pappas, Carol Gardner, Changjiang Guo, Andrew Gow.

**Project administration:** Kathleen Black, Debra Laskin, Howard Kipen, Michael Falvo.

**Resources:** Carol Gardner, Kathleen Black, Michael Falvo, Andrew Gow.

**Supervision:** Kathleen Black, Debra Laskin, Howard Kipen, Anays Sotolongo, Michael Falvo, Andrew Gow.

**Validation:** Gregory Pappas, Andrew Gow.

**Visualization:** Gregory Pappas.

**Writing – original draft:** Gregory Pappas, Michael Falvo, Andrew Gow.

**Writing – review & editing:** Gregory Pappas, Michael Falvo, Andrew Gow.

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
