## [Decision Letter · Decision Letter 0]

23 Apr 2025

Monocyte clusters suggestive of a chronic inflammatory phenotype are associated with reduced endothelial function in Veterans with respiratory symptoms

PLOS ONE

Dear Dr. Pappas,

Thank you for submitting your manuscript to PLOS ONE. After careful consideration, we feel that it has merit but does not fully meet PLOS ONE’s publication criteria as it currently stands. Therefore, we invite you to submit a revised version of the manuscript that addresses the points raised during the review process.

We look forward to receiving your revised manuscript.

Kind regards,

Tomasz W. Kaminski

Academic Editor

PLOS ONE

Journal Requirements:

5. Please amend the manuscript submission data (via Edit Submission) to include author Carol Gardner, Changjiang (C.J.) Guo , Kathleen Black , Raymond Rancourt, Debra Laskin , Howard Kipen , Anays Sotolongo, Michael Falvo and Andrew Gow.

Additional Editor Comments:

Dear Authors,

Thank you for the opportunity to handle your manuscript.Your work addresses an important clinical question and presents interesting findings that are appropriate for the readership of PLOS ONE.

While there are several points raised by the reviewers that require clarification or additional detail, I believe these revisions are manageable. Addressing these comments carefully will significantly improve the clarity, rigor and overall impact of your manuscript.

I encourage you to thoughtfully revise the manuscript in line with the reviewers’ suggestions and to provide clear responses to each point. With these revisions, your work has the potential to make a valuable contribution to the literature.

Wishing you all the best in the revision process.

Sincerely,

Tomasz W Kaminski

Reviewers' comments:

Reviewer's Responses to Questions

**Comments to the Author**

1. Is the manuscript technically sound, and do the data support the conclusions?

Reviewer #1: No

Reviewer #2: Yes

2. Has the statistical analysis been performed appropriately and rigorously?

Reviewer #1: No

Reviewer #2: Yes

3. Have the authors made all data underlying the findings in their manuscript fully available?

Reviewer #1: Yes

Reviewer #2: Yes

4. Is the manuscript presented in an intelligible fashion and written in standard English?

Reviewer #1: Yes

Reviewer #2: Yes

Reviewer #1: General comments: Although the manuscript includes some novel associations, such as the associations between lymphocyte clusters and flow mediated dilatation and oscillometry findings, the analysis of the other questionnaire-based data includes the analysis of many closely related questions (multiple categories of similar symptoms, for example). The also talk about drug development in the absence a specific disease is very premature.

Abstract: The phrase “Exposure to airborne hazards during deployment has been linked to persistent respiratory symptoms” is true, but the comment regarding vascular dysfunction is not common knowledge and is the basis of the current study. I also do not find the statement “Evidence suggests that circulating monocytes, key mediators of innate immune responses, contribute to these pathologic responses” in reference to respiratory symptoms is factual. I suggest rewriting the abstract to be more circumspect, focusing on the role of various environmental exposures on the immune system, and hypothesizing that such exposures may persist after exposure. This Veterans were studied years after exposure.

I also believe that finding association does not mean causality, and the statement “These results highlight the potential of monocyte-targeted interventions to mitigate chronic inflammation and downstream health effects in veterans with deployment-related exposures” is not warranted. We do not know the directionality of the associations they describe with inflammation, as many other cells and mediators are involved in the amplification of inflammation. The mention of therapy targeting monocyte activation is premature.

The last sentence is true of most author’s wishes “Further research is warranted, and should comment more on the significance of the research.

These edits should be included in the discussion text

Introduction:

Line 54. What does “and its associated diversity” mean in regards to particulate matter.

Lines 56-58: “Mechanisms underlying post-deployment respiratory symptoms and conditions include activation of cell-signaling pathways in responsive cells leading to pro-inflammatory responses that occurs secondary to inhalation of PM (5,6). This also appears in the introduction, and needs editing, as noted in my previous comments.

Line 72: “and increased self-reported dyspnea”. Note that dyspnea is multifactorial, and may not always reflect vascular health.

Lines 73-75: “Understanding the inflammatory profiles of monocytes in this population could provide insights into both the pathogenesis of deployment-related respiratory disease and associated health outcomes, suggesting future approaches for targeted interventions.” Is premature, as the assessment on monocyte activation in the setting of inflammation may be the consequence of other inflammatory pathways, and needs additional linkage to disease

Methods:

1. Were Veterans with cancer, receiving steroids/other immunomodulating drugs, or with chronic inflammatory conditions excluded?

2. Could past heavy smoking, in terms of having chronic inflammation in the lungs, in the absence of clinical disease, affect monocyte immunology? How was this addressed in the analysis?

3. Did all subjects have land-based deployments?

3. More information regarding spirometry procedures should be included. What predicted values were used for oscillometry?

4. Suggest including detail on the MMRC questions in a supplement.

5. More information regarding respiratory oscillometry (a more current description of what they did) is needed. What device was used?

6. More information on the protocols to collect and store monocytes are needed. How were batch effects handled in the 84 subjects?

7. Shouldn’t time from deployment be calculated from time since last deployment, not the first one?

8. The exposure questions require some reference, as is not clear what burn pit duties means?

Results: Some of the groups/cells compared are tiny (like the race categories?) Why assess race? There were too many comparisons, many assessing similar outcomes (such as multiple dyspnea categories, multiple deployment related categories). Time since initial deployment was “significant” …. but how does one interpret this, as all categories of mean years since first deployment are quite similar?

Discussion:

Lines 381-388: Given the heterogeneity of diagnosis in this population, I am concerned about making broad statements regarding the etiology of disease in this population, particularly with a small sample size. Even though they have no control group, there were participants without symptoms. What will the treatment trial focus on? There is no comment about the lack of association with symptoms or exposure…yet they talk of treatment?

Reviewer #2: Dear Author,

Your manuscript addresses a significant and often overlooked intersection between immunology, vascular biology, and military occupational health. The findings are innovative, and the methodology is sound. Your approach to using non-biased clustering of monocyte markers provides new perspectives on the chronic inflammatory mechanisms in post-deployment syndromes.

There are a few areas where further clarification or elaboration would enhance the manuscript:

• Control Group: The absence of a non-exposed or healthy control population is noted. Please consider discussing how this limitation affects the interpretation of your findings, and how future studies might address this, perhaps by including non-deployed veterans or matched civilian populations.

• Sample Size in Functional Measures: The subset of participants who completed the FMD/NMD testing (n = 48) is relatively small. It would be helpful to clarify whether these individuals are representative of the larger cohort, and to note any demographic or clinical differences between those who did and did not complete these measures.

• Temporal Disconnect: The immune data were collected long after deployment. While this is valuable in reflecting long-term effects, it also raises questions about immune memory versus ongoing inflammation. Could longitudinal immune profiling or additional biomarkers help distinguish between these?

• Data Availability: Ensuring that the individual-level data used for PCA and clustering are placed in a publicly accessible repository or provided in supplementary files would greatly facilitate reproducibility.

Your study is strong and innovative, making a notable contribution to the literature. However, it could benefit from a more detailed discussion of its limitations, particularly regarding the lack of control data and the variability in sample size for sub-analyses. Clarifying the data availability would further strengthen the manuscript's impact and scientific rigour. I look forward to reviewing the revised version.

**Do you want your identity to be public for this peer review?** For information about this choice, including consent withdrawal, please see our Privacy Policy

Reviewer #1: No

Reviewer #2: No

---

## [Author Response · Author response to Decision Letter 1]

16 Jul 2025

submitted as attachment "Response to Reviewers"

---

## [Decision Letter · Decision Letter 1]

26 Aug 2025

Dear Dr. Pappas,

Thank you for submitting your manuscript to PLOS ONE. After careful consideration, we feel that it has merit but does not fully meet PLOS ONE’s publication criteria as it currently stands. Therefore, we invite you to submit a revised version of the manuscript that addresses the points raised during the review process.

We look forward to receiving your revised manuscript.

Kind regards,

Tomasz W. Kaminski

Academic Editor

PLOS ONE

Journal Requirements:

Additional Editor Comments:

Dear Authors,

Thank you for your revision. Reviewers are satisfied overall but has requested a few additional clarifications. Please address Reviewer 1’s latest comments in your next, minor revision.

Best regards,

Tomasz W Kaminski

Reviewers' comments:

Reviewer's Responses to Questions

**Comments to the Author**

Reviewer #1: (No Response)

Reviewer #2: All comments have been addressed

2. Is the manuscript technically sound, and do the data support the conclusions?

Reviewer #1: Yes

Reviewer #2: (No Response)

3. Has the statistical analysis been performed appropriately and rigorously?

Reviewer #1: Yes

Reviewer #2: (No Response)

4. Have the authors made all data underlying the findings in their manuscript fully available?

Reviewer #1: Yes

Reviewer #2: (No Response)

5. Is the manuscript presented in an intelligible fashion and written in standard English?

Reviewer #1: Yes

Reviewer #2: (No Response)

Reviewer #1: The authors have comprehensively addressed the previous questions raised. However, I believe the results can be strengthened further by providing additional detail.

1. A major finding is differences in R5 %-predicted--

Unlike the spirometry predicted values, the predicted values for oscillometry are not robust, and the percent predicted values are not easily clinically interpretable. However, the raw values for oscillometry are more interpretable.

For example, values of R5 >5.1 cmH20/L/S, R5-R20 >1.0 cmH20/L/S, and Ax>10 cm H20/L are generally considered abnormal (Respir Med 2018; 139:106-9). I suggest presenting the raw values for oscillometry, presenting the raw values for R5-20 (not as a % of R5), including Fres, and analyzing the raw data as the primary analysis, but adjusting for age, BMI, height, and sex. The %-predicted analysis can go in the supplement. Was BMI included previously (as this can influence X5 and AX)? Including smoking using this approach adds another covariate in a modest data set-- so could do a sensitivity analysis excluding the smokers. This would would address if the smokers were influencing the increased R5.

2. Clarify if the GLI race specific (from 2012: white, black, other) or GLI global (race neutral) were used for spirometry?

What predicted values were used for TLC, RV?

3. Please clarify which analysis were adjusted for multiple comparisons (I assume the detailed monocyte data), not the comparison of pulmonary function or flow dilatation by cluster, in which one could argue that each comparison is an apriori independent test.

4. Please discuss the meaning of the differences in AX. Adding Fres to the data might help.

5. The paper is entitled "Monocyte clusters suggestive of a chronic inflammatory phenotype are associated with

reduced endothelial function in Veterans with respiratory symptoms", However, not all the participants had symptoms. In fact, the symptoms were not associated with monocyte cluster. Consider altering the title, and discussing the lack of association with symptoms.

Reviewer #2: (No Response)

**Do you want your identity to be public for this peer review?** For information about this choice, including consent withdrawal, please see our Privacy Policy

Reviewer #1: **Yes:** The authors have clarified there approach, but I believe the oscillometry data needs some additional detail. Please see my comments to the authors

Reviewer #2: **Yes:** JONATHAN AKWASI ADUSEI

---

## [Author Response · Author response to Decision Letter 2]

21 Nov 2025

Response to reviewers attached Response to Reviewers 2025-09-26

---

## [Editor Report · Decision Letter 2]

1 Dec 2025

Monocyte clusters suggestive of a chronic inflammatory phenotype are associated with reduced endothelial function in Veterans with respiratory symptoms.

PONE-D-25-03008R2

Dear Dr. Gow,

We’re pleased to inform you that your manuscript has been judged scientifically suitable for publication and will be formally accepted for publication once it meets all outstanding technical requirements.

Kind regards,

Tomasz W. Kaminski

Academic Editor

PLOS ONE
---

## [Editor Report · Acceptance letter]

PONE-D-25-03008R2

PLOS One

Dear Dr. Gow,

I'm pleased to inform you that your manuscript has been deemed suitable for publication in PLOS One. Congratulations! Your manuscript is now being handed over to our production team.

Kind regards,

on behalf of

Dr. Tomasz W. Kaminski

Academic Editor

PLOS One